# A Risk Treatment Strategy Model for Oil Pipeline Accidents Based on a Bayesian Decision Network Model

**DOI:** 10.3390/ijerph192013053

**Published:** 2022-10-11

**Authors:** Chao Zhang, Wan Wang, Fengjiao Xu, Yong Chen, Tingxin Qin

**Affiliations:** 1China National Institute of Standardization, Beijing 100191, China; 2Shenzhen Urban Public Safety and Technology Institute, Shenzhen 518000, China

**Keywords:** oil pipeline accident, risk analysis, risk evaluation, risk treatment strategy, Bayesian network, Bayesian decision network model

## Abstract

Risk treatment is an effective way to reduce the risk of oil pipeline accidents. Many risk analysis and treatment strategies and models have been established based on the event tree method, bow-tie method, Bayesian network method, and other methods. Considering the characteristics of the current models, a risk treatment strategy model for oil pipeline accidents based on Bayesian decision network (BDNs) is proposed in this paper. First, the quantitative analysis method used in the Event-Evolution-Bayesian model (EEB model) is used for risk analysis. Second, the consequence weights and initial event likelihoods are added to the risk analysis model, and the integrated risk is obtained. Third, the risk treatment strategy model is established to achieve acceptable risk with optimal resources. The risk treatment options are added to the Bayesian network (BN) risk analysis model as the decision nodes and utility nodes. In this approach, the BN risk analysis model can be transformed into a risk treatment model based on BDNs. Compared to other models, this model can not only identify the risk factors comprehensively and illustrate the incident evolution process clearly, but also can support diverse risk treatment strategies for specific cases, such as to reduce the integrated risk to meet acceptable criterion or to balance the benefit and cost of an initiative. Furthermore, the risk treatment strategy can be updated as the risk context changes.

## 1. Introduction

Oil pipelines are among the most important parts of energy utility systems. However, pipeline accidents may result in catastrophic consequences [1], including injury and death, economic loss, and environmental pollution. Many risk factors may lead to or influence oil pipeline accidents during the incident evolution process [2,3,4,5]. Some risk factors can be controlled with a risk treatment strategy. However, there are some disadvantages of the risk treatment strategy models, such as the comprehensives of risk factors, illustration of incident evolution, and effectiveness for a specified risk management purpose.

The purpose of risk treatment is to select and implement the best option to address risk. The best risk treatment options should be identified by balancing the potential benefits in relation to the associated costs, labor requirements, or other disadvantages of implementation. Based on the current risk analysis models, two additional steps are needed to fully implement risk treatment:Risk evaluation: evaluate the risk considering the likelihood and consequences to obtain the integrated risk, and compare the result with the acceptable risk criterion to determine whether it is acceptable.Risk treatment: analyze the potential benefits and costs of risk factors that can be controlled, and propose a risk treatment strategy.

Some conventional methods can address the issues associated with this type of risk. The fault tree analysis method [6], event tree method [7], and bow-tie method [8,9] use diagrams to illustrate the main processes that occur during the initial and subsequent stages of potential events. However, these methods simplify the risk factors using Boolean logic, and the relevant events are generally assumed to be mutually independent. It is difficult to use these methods to analyze the complex interactions among risk factors for oil pipeline accidents. Moreover, these methods are not suitable for dealing with the multistate and conditional probabilities of risk factors. Physical simulation methods based on computational fluid mechanics, such as oil leakage models [10], thermal radiation models for fires, and explosion overpressure models for vapor cloud explosions [11], can accurately and quantitatively simulate dynamic consequences [12,13]. Statistical methods based on cases [3,4,5] or expert experiences [14] can be used to calculate the occurrence probability of initial events [15] and the conditional probabilities of risk factors. The weights of diverse consequences can be obtained using the analytical hierarchy process method [16,17] or fuzzy logic theory, and these methods involve fuzzy comprehensive evaluation [18], intuitionistic fuzzy sets [19], and fuzzy inference systems [3,4,5,20]. Individual risk [21] and social risk [22,23,24,25,26] can be obtained by integrating physical simulation methods with the vulnerability of hazard-affected carriers in specific cases. Furthermore, the risk level [27,28] can be obtained using a risk matrix, and the subsequent risk management strategy can be obtained with genetic algorithms [29] or optimization methods. However, obtaining the risk treatment strategy for an oil pipeline accident requires illustrating the incident evolution process to comprehensively identify risk factors; quantitatively analyzing the corresponding effects, benefits, and costs; and comparing the risk treatment options. It is necessary to integrate the proper methods to address this complex problem.

Bayesian networks (BNs) are based on the Bayes rule. The main framework of a network involves nodes that represent risk factors and links that indicate the probability relationships between pairs of causal nodes. Then, the state probability of each node can be calculated using the Bayes rule. Furthermore, decision nodes in a Bayesian decision network (BDN) represent the discrete options that affect outcomes. In risk management problems, a decision node can be used as a risk treatment option to increase the potential benefits, while reducing cost and resource use. BDNs can display the potential benefits for various risk treatment decisions. Thus, we can use BDNs to obtain a risk treatment strategy for a given risk management purpose. Therefore, a BDN model combines a directed acyclic graph (DAG) and probability theory, and such models have been widely used in risk analysis [30,31] and risk treatment [32,33,34,35]. However, there are still some disadvantages of BDNs. Notably, it is difficult to comprehensively identify the elements related to different risk factors, and the incident evolution process is difficult to illustrate.

The EEB model was proposed for probabilistic analyses of oil pipeline network accidents [36]. It integrates an event tree, an incident evolution diagram, and a BN. First, the event tree is used to identify the main accident events and the consequences of the accident. It is useful to comprehensively identify the risk factors. Then, the incident evolution diagram (IDE) is used to illustrate the possible evolution paths and the corresponding key risk factors. It is useful to clarify the causal relationships between risk factors. Based on the available information, the BN is established for probabilistic analysis of the consequences. However, the EEB model cannot support the development of a risk treatment strategy because the costs and potential benefits of risk treatment options are not considered.

In this paper, we developed a BDN model to establish risk treatments for oil pipeline accidents. First, the risk analysis model used in the EEB model was applied to quantitatively analyze the risk of oil pipeline accidents. Then, we evaluated the integrated risk by setting the weight of each consequence and the likelihood of occurrence of initial events and comparing the results to the relevant acceptability criterion. Third, a risk treatment strategy was established by identifying the risk treatment options and analyzing by comparing their potential benefits and costs.

## 2. Methodology

An overview of the methodology proposed in this study is illustrated in Figure 1. There are three parts to this study:

Part 1—EEB model for risk analysis. An EEB model is obtained by integrating the event tree method, incident evolution diagram, and BN.

Part 2—Integrated risk evaluation for diverse consequences. The integrated risk evaluation is obtained by aggregating the magnitudes of consequences and their likelihoods according to the consequence weights.

Part 3—Risk treatment strategy based on BDN model. The risk treatment strategy is obtained by analyzing the risk treatment options according to the specified risk management purpose based on BDN model.

### 2.1. EEB Model for Risk Analysis

Generally, risk is defined as the effect of uncertainty on objectives. In an oil pipeline accident, there may be dozens of factors, each with inherent uncertainty, that affect consequent actions. These factors can be referred to as risk factors. Therefore, a risk analysis model for an oil pipeline accident should provide the following characteristics:Consider all risk factors comprehensively.Explicitly illustrate the causal relationships between risk factors.Quantitatively characterize the likelihood of event occurrence and the causal relationships of risk factors.Calculate the probabilities of consequences.

In addition, the incident evolution process and the relevant factors should be clearly expressed to explicitly and comprehensively identify risk factors.

The EEB model [36] was proposed based on the event tree method and the BN network method. The event tree integrated with incident evolution diagram can qualitatively express the incident evolution process and the event-triggering conditions. The BN method is appropriate for quantitative uncertainty analysis. There are four characteristics of the EEB model:All three sub-models, including the event tree, incident evolution diagram, and BN model, are comprehensively designed in reference to the relevant risk factors.Event tree models previously developed for typical incidents can be used to systematically analyze the evolution process.The incident evolution diagram is introduced to enhance the expression of the evolution process according to the relevant risk factors and their effect on the incident evolution path.A deployment rule for factor nodes in the BN is introduced to establish a BN model based on risk factor classification.

Although the event tree and BN are conventional tools, the incident evolution diagram is introduced in EEB model. It illustrates all incident evolution paths and five types of risk factors for the incident situation, the response mission, key environmental conditions, the emergency target, and consequences. The emergency target is a theoretical factor, and all other risk factors are related to real-world processes. It is important for different evolution paths to be considered, and we use dotted boxes to denote possible paths. A portion of an incident evolution diagram is shown in Figure 2.

In the EEB model, BN node deployment rules are used to assess oil pipeline accidents. Nodes are deployed from top to bottom according to the relevant causal relationships. The BN expresses risk information other than the incident evolution paths. To clarify the important causal relationships for different events, initial and secondary events are included in the second and third layers of the model, respectively. Moreover, trigger conditions are classified into two categories: initial event causes and secondary event causes. These causes are included in the first and second layers of the BN, respectively. A schematic of the BN node deployment rules for an oil pipeline accident is shown in Figure 3.

### 2.2. Integrated Risk Evaluation for Diverse Consequences

An oil pipeline accident can result in various consequences. The total level of risk is based on the aggregation of the magnitudes of these consequences and their likelihoods. The likelihood of each consequence can be obtained with the EEB model based on the assumption of an initial event occurring; the integrated risk level *R* is calculated as follows:(1)$$R=\left(\sum_{i}c_{i}\cdot p_{Ci}\cdot w_{i}\right)\cdot P$$
where *c_i_*, *p_Ci_*, and *w_i_* denote the typical value, probability, and weight of consequence *i* on the condition that the initial event occurred, and *P* denotes the probability of the initial event. The consequence weight *w_i_* could be obtained according to the emergency regulations, for example, the corresponding emergency plan.

The probability of each state occurring for each consequence can be obtained with the BN model. As each state is reflected by a numerical interval, Equation (1) can be transformed as follows:(2)$$R=\left(\sum_{i,j}\bar{c_{i,j}}\cdot p_{i,j}\cdot w_{i}\right)\cdot P$$
where $\bar{c_{i,j}}$ denotes the average value of the numerical interval, which is used as the typical value of state *j* for consequence *i*.

Then, the integrated risk level *R* can be compared to the acceptable risk criterion to determine whether the risk is acceptable or needs to be addressed.

### 2.3. Risk Treatment Strategy Based on BDN Model

The unacceptable risk should be decreased based on the risk criterion for a given event. It is necessary to select the most appropriate risk treatment options to balance their costs and potential benefits. For a real hazard, the best combination of risk treatment options, encompassed in a risk treatment strategy, should be effective enough to mitigate the risk with a reasonable amount of resources until it becomes acceptable.

Generally, risk treatment options can involve removing the risk source, changing the likelihood, changing the consequences, and so on. The risk treatment options for an existing oil pipeline system can be classified according to the risk factors, including the property of hazard, triggering condition that acts on the hazard and results in an incident, hazard-affected carriers as the object of hazardous effect, and the emergency response that can mitigate the consequences. The risk treatment options and the corresponding risk factors are shown in Table 1.

BDNs were used to develop a risk treatment strategy. Based on the BN node deployment rules for an oil pipeline accident, a risk treatment option was added in conjunction with each corresponding risk factor node, and the links denoting the causal relationships between the parent and child nodes were added. Then, the framework of the BDN model for the risk treatment strategy was established, as shown in Figure 4.

Based on BDN theory, two types of risk treatment strategies can be obtained. One combines risk treatment options, which can optimally balance the cost and potential benefits without considering the acceptable risk criterion. The second costs less than the first while reducing the risk based on the acceptable risk criterion. This model can be used as a safety planning tool for general conditions and as a risk treatment decision tool for specified hazards.

## 3. Risk Treatment Strategy Model for Oil Pipeline Network Accidents

### 3.1. EEB Model for Risk Analysis

A typical initial event that can cause an oil pipeline accident is oil leakage, which results from pipe cracking. This type of event is considered an initial event. Some subsequent events, such as fires and water pollution, can occur; they are denoted as secondary events. Additionally, vapor cloud explosions can occur in confined spaces, and such events are secondary events. Some issues may be easy to ignore but can result in catastrophic consequences; for example, the oil pipeline accident that occurred in Qingdao, Shandong Province, China, resulted in 62 deaths, 136 injuries, and more than CNY 75 million in damages [36].

In this study, the BN risk analysis model for an oil pipeline accident is established. The model includes 20 nodes, as shown in Table 2. They are classified into four types and described as follows.

#### 3.1.1. (I) Incident

(i.1)Occurrence time: fewer people are present late at night than at other times, and the emergency response is less effective then.(i.2)Accident area: the greater the number of people present, the more effective the emergency response; notably, the response is more effective in business districts than in remote areas.(i.3)Hazard properties.(i.31)Pipeline pressure and (i.32) pipeline capacity: more oil leaks occur in high-pressure and high-capacity pipelines than in low-pressure and low-capacity pipelines. The criteria for oil pipeline pressure may be different across countries. In China, the criteria are ≤4 MPa, 4~10 MPa, and ≥10 MPa for lower-, medium-, and high-pressure pipelines, respectively.(i.4)Initial event causes: an oil pipeline will crack under certain types of external interference. Corrosion is a frequent cause of pipeline failure. Design defects, including construction defects, technical defects, and material defects, are inherent causes of failure. Geological disasters can completely destroy oil pipelines. In addition, there are several other causes of initial events.(i.5)Secondary event causes.(i.51)Immediate ignition, (i.52) delayed ignition, and (i.53) confined space nearby are key environmental conditions associated with secondary events after oil has leaked from the pipe and onto the ground. Both immediate ignition and delayed ignition are causes of fires and explosions. The differences between these processes are the occurrence probabilities and the severities of the consequences. If confined spaces are located near the incident location, the oil may leak into them and explode as the oil concentration increases.(i.54)Water area nearby: this is a key environmental condition for secondary confined space explosions. The true and false states represent the presence or absence of water areas, respectively.(i.6)Initial event—oil leakage: oil leakage is the initial event that results if a pipeline cracks; such an event can occur at three qualitative scales: small, medium, and large.(i.7)Secondary events.(i.71)Secondary fires, (i.72) secondary vapor cloud explosion, and (i.73) secondary water pollution are three types of secondary events. Their states are based on three qualitative scales of small, medium, and large.

#### 3.1.2. (H) Hazard-Affected Carriers

(h.1)Threatened persons: this is the most important hazard-affected carrier and is related to the number of potential casualties. The state of this variable is based on different scales of population density.(h.2)Buildings and (h.3) infrastructure and lifelines: damage to these hazard-affected carriers may result in economic losses.

#### 3.1.3. (E) Emergency Response

(e)Emergency response: measures can be used to control the initial event and secondary events and reduce the subsequent consequences. The three response states are effective, general, and poor.

#### 3.1.4. (C) Consequences

(c.1)Casualties, (c.2) economic losses, and (c.3) environmental pollution are three direct consequences of an initial event and secondary events; each consists of four states that describe different quantitative scales.

All the factor nodes were deployed based on the rules shown in Figure 3. The occurrence probabilities and conditional probabilities of the factor nodes were based on comprehensive case studies and were further evaluated by expert judgment with weights using Dempster–Shafer evidence theory [37]. The BN risk analysis model for oil pipeline accidents is shown in Figure 5. It was produced with Netica (Norsys and Netica are trademarks of Norsys Software Corp. Copyright© 2022–2014 by Norsys Software Corp) software, which is a complete software package that includes Bayesian belief networks, decision nets, and influence diagrams.

### 3.2. Integrated Risk Evaluation for Diverse Consequences

For the risk evaluation, the consequence weights were set according to the emergency plan, the acceptable risk criterion was set based on the risk management requirements for the targeted hazard, and the initial event likelihood considered both the accident probability statistics and the oil pipeline properties.

The integrated risk was calculated with Equation (2), and the corresponding BN risk analysis model is shown in Figure 4. Three types of BN nodes, namely, the occurrence time (i.1), accident area (i.2) and hazard properties (i.3), and the probabilities of their states were set for the general situation involving an oil pipeline accident. The initial event was the oil leakage (i.6), which could result in some secondary events, such as fires (i.71), vapor cloud explosion (i.72), and water pollution (i.73). The triggering conditions included the initial event causes (i.4), immediate ignition (i.51), delayed ignition (i.52), confined space nearby (i.53), and water area nearby (i.54). There were three types of consequences: casualties (c.1), economic losses (c.2), and environmental pollution (c.3). The corresponding hazard-affected carriers, namely, threatened persons (h.1), buildings (h.2), and infrastructure and lifelines (h.3), were considered in the consequence layer. Each consequence is associated with diverse situations, which are denoted as *c_ij_*. The probabilities of the consequence states are shown in Figure 5 and are denoted as *p_ij_* in Equation (1).

The average value, denoted as $\bar{c_{ij}}$, is used to calculate the integrated risk *R*. Obtaining the average of the most serious state is difficult because the upper limit is infinite; notably, the corresponding likelihood is always extremely small, and a conservative value can be used. In this study, we use twice the average for the less serious state as the average for the most serious state. For example, the most serious state of casualties (*c*_1_) is “>30 persons” (*c*_14_), and the average value of *c*_13_ is 20. Thus, the number 20 × 2 = 40 is used as $\bar{c_{14}}$.

Some regulations are the basis for emergency management. In this paper, the consequence weights, denoted as set *w_ij_*, are defined based on the incident. The dangerous chemical accidents are classified into four levels in China, as shown in Table 3. The consequence weights and the level criteria are according to regulation of China, namely National Emergency Response Plan for Industrial Accident. Considering the cost of water pollution treatment, which may be CNY 10 million for a 1 km^2^ polluted water area, the weights of casualties, economic losses (million CNY), and water pollution (1 km^2^) are defined as 5, 1, and 10, corresponding to w_1_, w_2_, and w_3_, respectively.

The incident probability *P* is the likelihood of the target oil pipeline experiencing an accident. Focus is placed on the oil pipelines in an industrial park, and *P* is assumed to be 0.01/year. In this situation, Equation (2) can be transformed to:(3)$$R=\left[\begin{array}{l} \left(2.5p_{11}+7.5p_{12}+20p_{13}+40p_{14}\right)\times 5\;+\left(5p_{21}+30p_{22}+75p_{23}+150p_{24}\right)\times 1\;\\ +\left(0.5p_{31}+5.5p_{32}+30p_{33}+60p_{34}\right)\times 10 \end{array}\right]\times 0.01$$

### 3.3. Risk Treatment Strategy Based on the BDN Model

Four types of risk treatments are shown in Figure 4: changes to the hazard properties, triggering conditions of initial and secondary events, emergency response capability, and hazard-affected carriers. First, the triggering condition of the initial event is determined from statistical data. Then, changing the properties of hazards and the hazard-affected carriers is usually very costly. These types of options should only be considered if nothing else is effective enough to reduce the integrated risk to meet the acceptable risk criterion. Therefore, two other types of risk treatment options, namely, controlling triggering conditions of secondary events and improving emergency response capabilities, are generally considered firstly.

It is easier to control delayed ignition (i.52) than immediate ignition (i.51) after an oil leakage event occurs. The emergency mission in the emergency plan can be selected, and teams and resources can be assigned to control the event. It was supposed that, for the studied mission, it cost CNY 0.01 million to reduce the probability of occurrence from 30% to 0.01%. The secondary condition confined space nearby (i.53) is a key environmental condition that can be eliminated before or after the initial event. For a specified scenario with a nearby water area, it was assumed to cost CNY 0.1 million to eliminate this triggering condition. We could also improve the emergency response capability by implementing an emergency plan, performing training, and allocating emergency resources. It was assumed that the option to improve the emergency response capability costs CNY 0.1 million, while an additional level of improvement could be achieved for CNY 0.2 million. Therefore, three decision nodes, namely, the Treatment_Control Ignition, Treatment_Eliminate Water Area Nearby, and Treatment_Improve Capability nodes, and the corresponding utility nodes, such as the TotalLoss node indicating the integrated consequences, were added to the BN risk analysis model. The risk treatment model based on a BDN is shown in Figure 6. The utility node TotalLoss will dynamically indicate the integrated consequences as the states of decision node change.

## 4. Results and Discussions

### 4.1. Risk Evaluation without Risk Treatment Options

A specified oil pipeline scenario is often uncertain. To comprehensively analyze the effects of risk treatment options, the confined space nearby node was assumed to exist. Then, the probabilities of consequence states were calculated with the BN risk analysis model, shown as Figure 5, with the probability of “true” for the confined space nearby node ranging from 80% to 100%. The corresponding integrated risk *R* = 1.56 was obtained with Equation (3). Notably, CNY 1.56 million per year was determined to be the integrated risk for these oil transportation pipelines in the target industrial park. If the acceptable risk criterion, denoted as *R_c_*, was CNY 1.40 million per year, this scenario would be unacceptable as is, and some risk treatment options would need to be implemented based on the BN risk treatment model.

### 4.2. The Risk Treatment Strategy That Meets the Acceptable Risk Criterion

The risk treatment strategies involve three options: controlling delayed ignition, eliminating confined space nearby and improving the emergency response capability. All combinations of these three options should be investigated, and the corresponding results, including the integrated risk *R*, total cost of risk treatment options total_cost_, and their sum, denoted as *R* + total_cost_, can be obtained with the BN risk treatment model for oil pipeline accidents, shown as Figure 6. There were 12 risk treatment options, and the corresponding results are shown in Table 4.

The acceptable risk criterion (*R*_c_) value was assumed to be 1.4. To meet the acceptable risk criterion, the risk treatment strategy that minimized total_cost_ in the case of *R* < *R_c_* was selected. The total_cost_ and *R* values for each risk treatment strategy are shown in Figure 7. All the strategies with *R* values below the threshold line *R_c_* could be chosen. Among them, the minimum value of total_cost_ is 0.1. The corresponding strategies are strategies No. 3, No. 5, No. 9, and No. 11. As the lowest *R* value was 1.27 in strategy 5, strategy 5 was selected as the best risk treatment strategy to meet the acceptable risk criterion. The delayed ignition conditions must be controlled and the emergency response must be improved in this strategy.

### 4.3. The Risk Treatment Strategy That Best Balances Cost and Potential Benefits

With the acceptable risk criterion, we can obtain a risk treatment strategy that best balances the cost and potential benefits, regardless of the acceptable risk criterion. The values of total_cost_ and *R* and their sum are shown in Figure 8. The minimum sum value, denoted as (total_cost_+*R*)_min_, is 1.20. This result indicates that the best strategies in terms of balance are No. 1 and No. 7. If a higher risk level was allowable and the emergency response cost was restricted, the best strategy would be No. 9. In this case, the confined space nearby must be eliminated, and the emergency response capability must be improved. Otherwise, strategy No. 1 should be implemented, in which case the delayed ignition condition should be controlled in addition to the options in strategy No. 7.

### 4.4. The Risk Treatment Strategy for a Specified Hazard

A risk treatment strategy could be obtained for a specified hazard. For a section of pipeline located in a business district, both the pressure and capacity were assumed to be moderate. The risk treatment options are shown in Table 5. It was best to decrease *R_c_* in the business district area. If *R_c_* was set to 1.2, the minimum value of total_cost_ was calculated as 0.2 for the No. 8 and No. 10 strategies, for which *R* is less than *R_c_*. Furthermore, the best risk option could be obtained based on *R*+total_cost_. Overall, the No. 10 risk treatment option is effective and only requires improving the emergency response capability without controlling delayed ignition or eliminating confined space nearby.

### 4.5. Risk Updating Coupled with Urbanization

As the risk context of a target pipeline may change continuously, the risk should be updated, and the risk treatment strategy should be simultaneously adjusted. A common situation is urbanization. In this process, the population density changes, and the state probability of threatened persons nodes should be updated. Table 6 and Table 7 show the conditional probability tables for threatened persons before and after urbanization. Moreover, the risk treatment options and the corresponding model results obviously change, as shown in Table 8. For the risk treatment options with the lowest *R* values, the best choice was No. 7, which involved eliminating confined space nearby and effectively improving the emergency response. If this risk treatment option was implemented, the minimum values of total_cost_ and *R* + total_cost_ would be obtained. Therefore, the risk treatment strategy was updated as the risk context changed.

## 5. Conclusions

In this study, a risk treatment strategy model for oil pipeline accidents was developed based on a BDN model. The EEB model, which is an integration of event tree, incident evolution diagram, and BN, was used for risk analysis. The consequence weights, initial event likelihoods, and acceptable risk criteria were used to obtain the integrated risk and determine whether it was acceptable. For unacceptable risk, the risk treatment model was used to develop a relevant treatment strategy. The effectiveness and cost of different risk treatment options were considered using the decision and utility nodes in the BDN model, respectively, and the risk treatment strategy model was developed. This model can suppose the risk treatment strategy for a specified hazard and be updated with the context changing resulting from urbanization.

The risk treatment model proposed in this paper can comprehensively consider various risk factors, evaluate the integrated risk considering diverse consequences, and be used to select a specific risk treatment strategy. Therefore, the risk levels calculated with this model are considered accurate and systematic. Moreover, the proposed model can be used for diverse risk management purposes, such as to reduce risk to an acceptable criterion or to best balance the cost and potential benefits of treatment. Additionally, this model not only supports the selection of the best risk treatment strategy for a specific hazard, but also can be used to update risk treatment strategies as the risk context changes.

Some additional work could be performed in the future based on this model. First, a function relationship could be used instead of a conditional probability relationship to improve the quantitative analysis accuracy. Second, the emergency response capability could be classified into several detailed levels. Notably, a natural state node would need to be established for each emergency response level, and the dependence relationships among nodes could be assessed. In this way, we could enhance the emergency response capability in areas by developing targeted plans and specific missions.

This paper established the risk treatment model from the perspective of risk. It limits the risk treatment options. For this study, tasks such as changing the hazard properties, hazard-affected carrier properties, and general circumstantial conditions have not been considered in risk treatment due to the high cost. Currently, the resilience concept has been widely used in infrastructure management [38]. The risk treatment strategy model could potentially be transformed into a resilience strategy model by considering more risk treatment options and changing the target utility to systematically maximize different capabilities.

## Figures and Tables

**Figure 1 ijerph-19-13053-f001:**
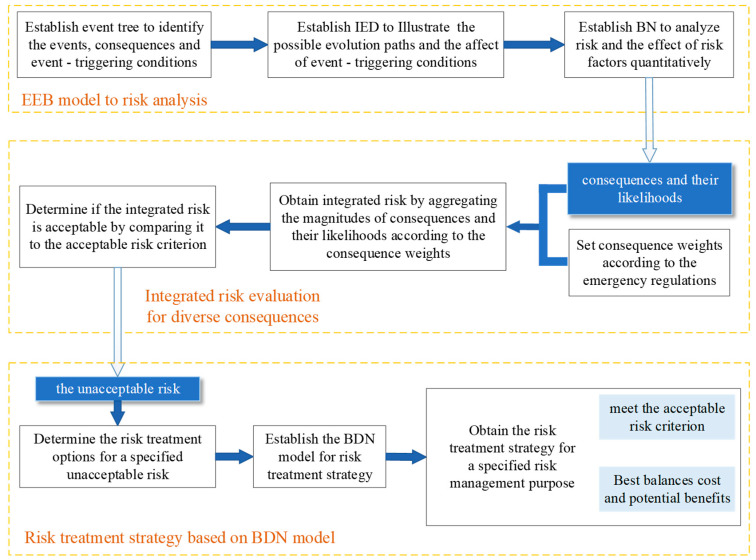
Schematic diagram of risk treatment strategy model based on a BDN model.

**Figure 2 ijerph-19-13053-f002:**
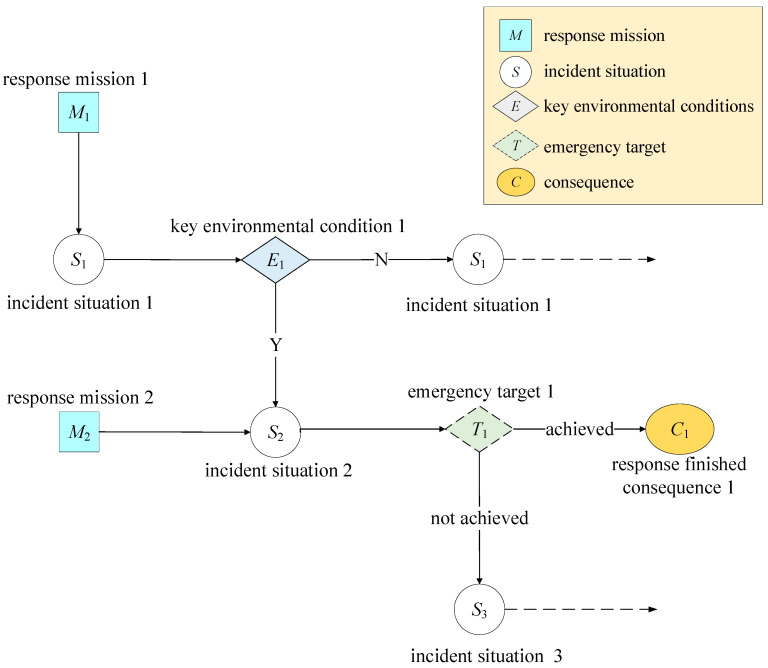
A portion of an incident evolution diagram.

**Figure 3 ijerph-19-13053-f003:**
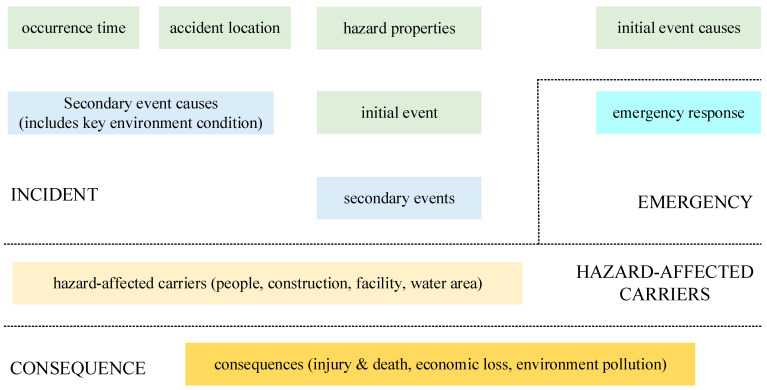
The BN node deployment rules for an oil pipeline accident.

**Figure 4 ijerph-19-13053-f004:**
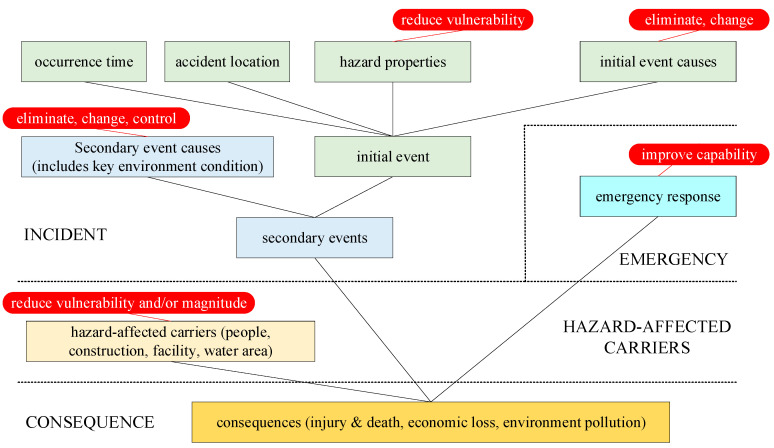
The framework of the BDN model for risk treatment strategy development.

**Figure 5 ijerph-19-13053-f005:**
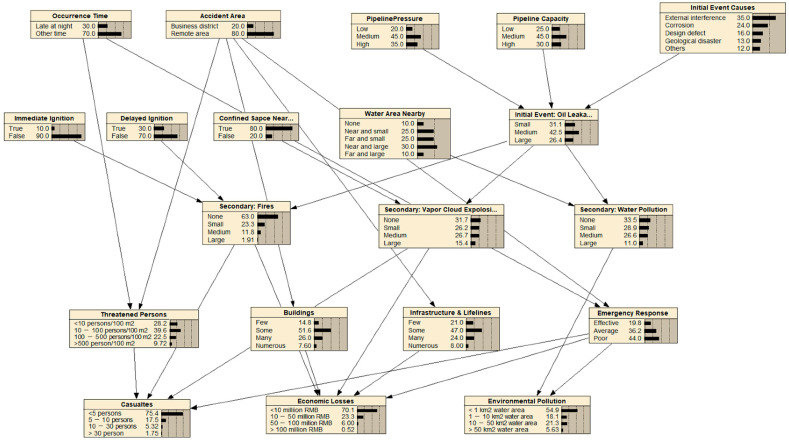
The BN risk analysis model for oil pipeline accidents.

**Figure 6 ijerph-19-13053-f006:**
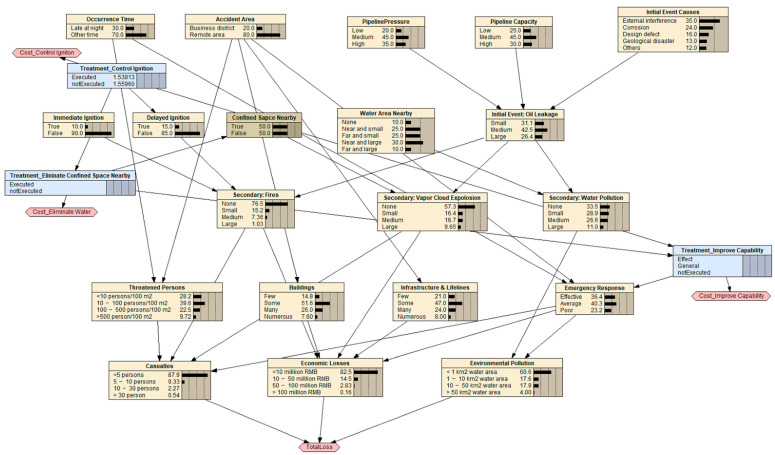
The BDN risk treatment model for oil pipeline accidents.

**Figure 7 ijerph-19-13053-f007:**
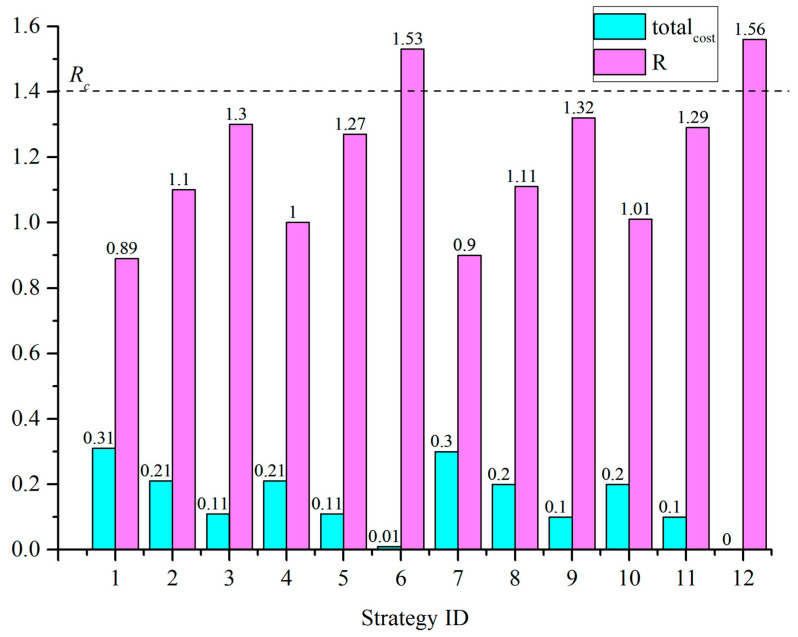
The total cost and *R* for each risk treatment strategy.

**Figure 8 ijerph-19-13053-f008:**
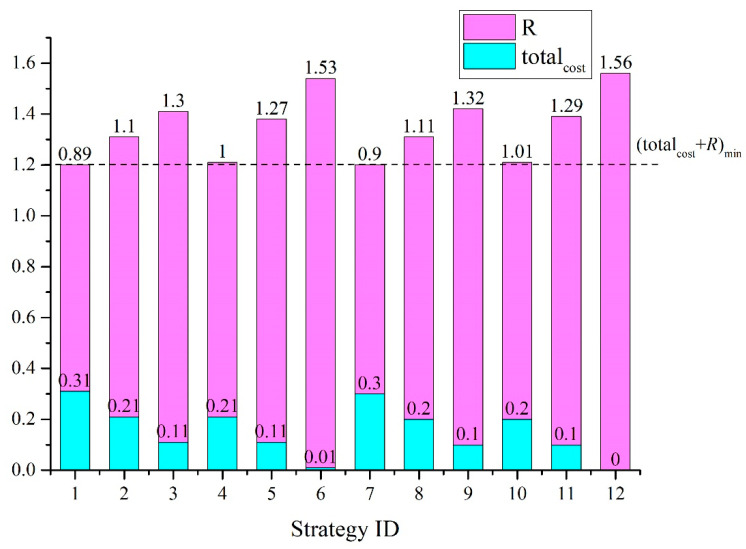
The total_cost_ and *R* values and their sum for each risk treatment strategy.

**Table 1 ijerph-19-13053-t001:** Risk treatment options for oil pipeline accidents.

Risk Factor	Risk Treatment Option
hazard property	reduce vulnerability
triggering condition	eliminate or change the condition before an event occurs, control triggering conditions to avoid secondary events after the initial event occurs
emergency response	improve the overall or targeted emergency response capability
hazard-affected carriers	reduce vulnerability, reduce their magnitude

**Table 2 ijerph-19-13053-t002:** The nodes in a BN risk analysis model for oil pipeline accidents.

Bayesian Nodes	States of Nodes
(i.1) occurrence time	① late at night; ② other time
(i.2) accident area	① business district; ② remote area
(i.31) pipeline pressure	① low, ② medium, or ③ high
(i.32) pipeline capacity	① low, ② medium, or ③ high
(i.4) the initial event causes	① external interference, ② corrosion, ③ design defect, ④ geological disaster, or ⑤ others
(i.51) immediate ignition	① true; ② false
(i.52) delayed ignition	① true; ② false
(i.53) confined space nearby	① true; ② false
(i.54) water area nearby	① none, ② near and small, ③ far and small, ④ near and large, or ⑤ far and large
(i.6) initial event: oil leakage	① small, ② medium, or ③ large
(i.71) secondary: fires	① none, ② small, ③ medium, or ④ large
(i.72) secondary: vapor cloud explosion	① none, ② small, ③ medium, or ④ large
(i.73) secondary: water pollution	① none, ② small, ③ medium, or ④ large
(h.1) threatened persons	① <10 persons/100 m^2^, ② 10–100 persons/100 m^2^, ③ 100–500 persons/100 m^2^, or ④ >500 persons/100 m^2^
(h.2) buildings	① few, ② some, ③ many, or ④ numerous
(h.3) infrastructure and lifelines	① few, ② some, ③ many, or ④ numerous
(e) emergency response	① effective, ② average, or ③ poor
(c.1) casualties	① <5 persons, ② 5–10 persons, ③ 10–30 persons, or ④ >30 persons
(c.2) economic losses	① <10 million RMB, ② 10–50 million RMB, ③ 50–100 million RMB, or ④ >100 million RMB
(c.3) environmental pollution	① <1 km^2^ water area, ② 1–10 km^2^ water area, ③ 10–50 km^2^ water area, or ④ >50 km^2^ water area

**Table 3 ijerph-19-13053-t003:** The levels and criteria for dangerous chemical accidents in China.

Incident Level	Casualty	Economic Loss (Million CNY)
I	≥30	≥100
II	[10,30)	[50,100)
III	[3,10)	[10,50)
IV	<3	[0.5,10)

**Table 4 ijerph-19-13053-t004:** Risk treatment options and their results.

Strategy ID	Risk Treatment Options			*R* + Total_cost_	Total_cost_	*R*
	Control Delayed Ignition	Eliminate Confined Space Nearby	Improve Response Capability			
1	Executed	Executed	Effective	1.20	0.31	0.89
2	Executed	Executed	General	1.31	0.21	1.10
3	Executed	Executed	Not executed	1.41	0.11	1.30
4	Executed	Not executed	Effective	1.21	0.21	1.00
5	Executed	Not executed	General	1.38	0.11	1.27
6	Executed	Not executed	Not executed	1.54	0.01	1.53
7	Not executed	Executed	Effective	1.20	0.30	0.90
8	Not executed	Executed	General	1.31	0.20	1.11
9	Not executed	Executed	Not executed	1.42	0.10	1.32
10	Not executed	Not executed	Effective	1.21	0.20	1.01
11	Not executed	Not executed	General	1.39	0.10	1.29
12	Not executed	Not executed	Not executed	1.56	0	1.56

**Table 5 ijerph-19-13053-t005:** Risk treatment options and their results for a specified hazard.

Strategy ID	Risk Treatment Options			*R* + Total_cost_	Total_cost_	*R*
	Control Delayed Ignition	Eliminate Confined Space Nearby	Improve Response Capability			
1	Executed	Executed	Effective	1.44	0.31	1.13
2	Executed	Executed	General	1.38	0.21	1.17
3	Executed	Executed	Not executed	1.34	0.11	1.23
4	Executed	Not executed	Effective	1.36	0.21	1.15
5	Executed	Not executed	General	1.36	0.11	1.25
6	Executed	Not executed	Not executed	1.39	0.01	1.38
7	Not executed	Executed	Effective	1.43	0.3	1.13
8	Not executed	Executed	General	1.38	0.2	1.18
9	Not executed	Executed	Not executed	1.34	0.1	1.24
10	Not executed	Not executed	Effective	1.36	0.2	1.16
11	Not executed	Not executed	General	1.36	0.1	1.26
12	Not executed	Not executed	Not executed	1.4	0	1.40

**Table 6 ijerph-19-13053-t006:** Conditional probability table for threatened persons before urbanization.

Occurrence Time	Accident Area	<10 Persons/100 m^2^	10–100 Persons/100 m^2^	100–500 Persons/100 m^2^	>500 Persons/100 m^2^
Late at night	Business district	54	40	5	1
Late at night	Remote area	80	18	2	0
Other time	Business district	1	15	35	49
Other time	Remote area	10	55	30	5

**Table 7 ijerph-19-13053-t007:** Conditional probability table for threatened persons after urbanization.

Occurrence Time	Accident Area	<10 Persons/100 m^2^	10–100 Persons/100 m^2^	100–500 Persons/100 m^2^	>500 Persons/100 m^2^
Late at night	Business district	30	50	15	5
Late at night	Remote area	65	25	7	3
Other time	Business district	1	6	38	55
Other time	Remote area	3	49	38	10

**Table 8 ijerph-19-13053-t008:** Risk treatment options and corresponding modeling results after urbanization.

Strategy ID	Risk Treatment Options			*R* + Total_cost_	Total_cost_	*R*
	Control Delayed Ignition	Eliminate Confined Space Nearby	Improve Response Capability			
1	Executed	Executed	Effective	1.51	0.31	1.20
2	Executed	Executed	General	1.52	0.21	1.31
3	Executed	Executed	Not executed	1.52	0.11	1.41
4	Executed	Not executed	Effective	1.43	0.21	1.22
5	Executed	Not executed	General	1.5	0.11	1.39
6	Executed	Not executed	Not executed	1.57	0.01	1.56
7	Not executed	Executed	Effective	1.5	0.3	1.20
8	Not executed	Executed	General	1.51	0.2	1.31
9	Not executed	Executed	Not executed	1.52	0.1	1.42
10	Not executed	Not executed	Effective	1.42	0.2	1.22
11	Not executed	Not executed	General	1.5	0.1	1.40
12	Not executed	Not executed	Not executed	1.58	0	1.58

## Data Availability

Not applicable.

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
