# Peer review of "A Risk Treatment Strategy Model for Oil Pipeline Accidents Based on a Bayesian Decision Network Model"

_ijerph, 2022, doi:10.3390/ijerph192013053_

Round 1

Reviewer 1 Report

I would like to thank the authors for their hard work and the interesting topic that they provided. I have some comments that should be considered in the updated version of the manuscript: 

1- Abstract (BN) should be wirtten as full name.

2- Introduction: Line38, Line 52: references are needed.

3- Methodology: it is not clear the description of the methodology and the rationale for selecting it! Also, it should be clear how many participants are involved in developing the model. How many participants are involved in validating it? collecting the needed data and assessments? and the criteria of selecting those participants? 

4- Table 3: the source of this data should be cited!

5- Results: the source of selecting the three options of the strategies should be mentioned and why? 

6- conclusions: based on the above changes and modifications, this section should be updated. The methodology should be mentioned and positive and negative experiences should be presented in short.  

Reviewer 2 Report

Thank you very much for giving me the opportunity to review this MS. This study present a new way for risk assessment and risk treatment of oil pipelines. The study is interesting but there are some points to be revised by authors:

- There are some typo errors throughout the MS,

- the Kent Muhlbauer is a very popular technique for pipeline risk assessment, what are the advantages of this method over the Kent method, 

- Bayesian decision graphs are slightly different from Bayesian networks, authors should explain the differences in more details. Moreover, authors must explain how new nodes in a decision graph works.

- in Table 1 risk treatment options are explained but the source of the table is not provided. It is recommended to add proper references to this section. 

- Authors repeated used immediate ignition and ignition, I think using immediate ignition and delayed ignition is more understandable. 

- in Table 2, more elaborated definitions of states are required. For example, which pressure is regarded as low or what was the criterion for categorizing pipeline capacity. 

- in the Bayesian network model, as immediate ignition and delayed ignition are mutually exclusive, they can be merged into one node with two states of immediate ignition and delayed ignition or one node with three states of immediate ignition, delayed ignition, and no ignition. 

- Crude oil is a heavy substance and its vaporization rate is commonly, is the VCE is a valid scenario? 

- in the Bayesian network, there is no link between ignition (delayed ignition) and VCE, while we need an ignition for VCE. I think the network must be revised in this regard. 

- in the Bayesian network, there is no link between VCE and building, while VCE can affect buildings. Authors should revise this issue. 

Reviewer 3 Report

This study presents a relatively new framework for accident analysis in oil pipeline. There are some issues which should be verify by authors:

a. The authors previously published an article titled " A probabilistic analysis model of oil pipeline accidents based on an integrated Event-Evolution-Bayesian (EEB) model ". What is the difference or innovation of this article compared with the previous one.

 b. The background of the study and the consequences of the oil pipeline accident are not very clear in the summary and introduction sections and need more explanation.

 c. The authors said “methods that support the development of risk treatment strategies for oil pipeline accidents are relatively inadequate.”, which is a statement to be considered. At this stage, the research on scenario evolution is hot, and the research results on emergency response mechanisms based on scenario models have existed for a long time in emergency response. The strategy feedback proposed by the authors is similar to the game theory in the scenario model, which identifies the good and bad regulation strategies by the change of the set objective function.

 d. There is a grammatical error in the first 3. of methodology, and the author should check the text well to see if there are still grammatical problems.

 e. It should have a roadmap of the overall research idea. A comparison of the roadmap with the structure of the article can identify the standardization and feasibility of the methodology and can make the overall expression of the article clearer.

 f. The authors use accident risk and strategy cost as criteria for judging the goodness of the corresponding behavior. So how do the authors consider the definition of accident risk probability and consequence and strategy cost. How do the authors consider the risk of each event for different scenarios with time variation and strategy feedback. Furthermore, how do the authors quantify them.

 g. How do the authors consider the extent to which different perspectives in risk consequences, such as human fatalities, economic losses or costs, contribute to the combined risk.

Round 2

Reviewer 1 Report

Thanks for your hard work and interesting paper. 

Reviewer 3 Report

All comments have been responded, and the paper can be accepted.